# Prescriptions of Antipsychotics in Younger and Older Geriatric Patients with Polypharmacy, Their Safety, and the Impact of a Pharmaceutical-Medical Dialogue on Antipsychotic Use

**DOI:** 10.3390/biomedicines10123127

**Published:** 2022-12-04

**Authors:** Eva-Maria Gebauer, Albert Lukas

**Affiliations:** St. Martinus Hospital, 40219 Düsseldorf, Germany

**Keywords:** antipsychotics, polypharmacy, geriatric, prescription, safety, pharmacist

## Abstract

Geriatric patients are a particularly vulnerable and, at the same time, very heterogeneous group due to their multimorbidity and polypharmacy. Antipsychotics are often prescribed in their complex drug regimens, whereby the prescription of antipsychotics is not without controversy. To date, questions remain as to whether there are differences in the prescribing pattern, safety, and impact of a consultant pharmacist regarding antipsychotic use between younger and older geriatric patients in the heterogenic geriatric group. This monocentric study of 744 patients was based on the analysis of routine data collected from January 2018 to June 2020 in a geriatric department during a weekly pharmaceutical and medical consultation. The frequency of the prescription of antipsychotics in our study was 30.7%. Regarding antipsychotic safety and/or adverse drug reaction (ADR) antipsychotics, only a difference in terms of overuse in younger geriatric patients was found. The binary logistic regression analyses of geriatric patients with antipsychotics revealed that ADRs and drug–drug interactions (DDIs) were particularly related to the number of medications prescribed. The higher the number of prescribed drugs, the higher the risk of ADRs and DDIs. In 26.7% of geriatric patients on antipsychotics, the pharmacist made recommendations that were almost exclusively implemented by the physician, with no difference made between the two age groups. The prescriptions of antipsychotics in geriatric patients with polypharmacy, their safety, and the impact of a pharmaceutical-medical dialogue on the use of antipsychotics seem comparable between younger and older geriatric patients in the geriatric setting. Antipsychotics should always be critically considered and used cautiously, whereby a regular pharmaceutical-medical dialogue is recommended in geriatric settings.

## 1. Introduction

Geriatric patients represent a particularly vulnerable group of patients who, by definition, suffer from multimorbidity, a simultaneous occurrence of two or more diseases [1]. Multimorbidity, in turn, almost inevitably leads to polypharmacy resulting in sometimes very complex drug regimens [2]. These drug regimens include multi-target drugs such as antipsychotics [3]. Antipsychotics are used in schizophrenia [4], dementia [5,6], bipolar disorders [7], and other mental disorders [8]. Indeed, a patient may benefit from a variety of medications administered in an evidence-based manner while always taking into account the combination of all clinical conditions and potential interactions with other medications [9].

The prescription of antipsychotics is not without controversy as they increase the risk of adverse drug reactions (ADRs) [3,10,11] and side effects [10,12]. For example, antipsychotics are associated with increased and prolonged hospitalizations [10]. With their extrapyramidal side effects [8,13], they can lead to increased gait disturbances [13,14]. In addition, there is an increased risk of all-cause mortality associated with antipsychotic use in older people with dementia [15] and side effects such as anticholinergic side effects [16], weight gain [8], type II diabetes mellitus, and cardiovascular diseases [12,16,17,18]. Therefore, the general recommendation is that antipsychotics should be used as cautiously and judiciously as possible in older people [19,20,21].

In general, however, the risk of adverse drug reactions also increases with the number of medications prescribed which enhances the likelihood of drug-related adverse events, drug–drug interactions (DDIs), compliance failures, and medication management errors [22]. In addition, age-related physiological changes that affect the pharmacokinetics and dynamics of many medications must be considered [23]. This includes the special consideration of impaired renal and hepatic functions to identify medications potentially unsuitable for geriatric patients [24].

Geriatric patients with polypharmacy and antipsychotics, therefore, represent a high-risk group. So far, it is unclear whether older geriatric patients are prescribed antipsychotics to a comparable extent compared to younger geriatric patients and whether they are comparably safe in their use. Similarly, it is not known whether conducting a pharmacologic-medical dialogue in the form of a joint weekly visit contributes to improved prescribing behavior for antipsychotics in geriatric patients of different age groups.

Therefore, this study specifically aims to answer the following questions related to prescribing patterns, the safety of medications used, and the impact of a pharmaceutical medical consultation:

(1.) Is there a difference in the prescription pattern of antipsychotics in younger and older geriatric patients with polypharmacy?

(2.) Is there a difference in neuroleptic potency or type of antipsychotics used in the two age groups?

(3.) (a) Is there a difference in the adverse drug reactions (ADRs) of antipsychotics in younger and older geriatric patients with polypharmacy?

(b) Is there a correlation between age, gender, glomerular filtration efficiency, the number of prescribed drugs, and the frequency of ADRs (and their sub-categories) in the overall age group studied and in younger and older geriatric patients?

(4.) Is there a difference with regard to the pharmaceutical recommendations given in the context of a pharmaceutical-medical consultation in younger and older geriatric patients with polypharmacy?

(5.) In general, what is the impact of a pharmaceutical-medical dialogue in the context of a joint weekly visit on the prescribing behavior of antipsychotics in geriatric patients of different age groups?

## 2. Materials and Methods

### 2.1. Study Design

The following evaluation is based on routine data collected from January 2018 to June 2020 during a weekly joint pharmaceutical-medical visit. The rounds were conducted jointly by a pharmacist and a geriatrician. In this consultation, clinically relevant medication problems relating to polypharmacy and multimorbidity in geriatric patients were discussed in detail between the geriatrician and pharmacist, and any necessary consequences were drawn. The clarification of the medical indication and drug selection and dosage, and application were part of this discussion, as well as the interactions of the drugs used, side effects, and possible documentation errors.

Before the start of the weekly visit, a detailed medication analysis was performed by the pharmacist for all patients with the aid of the patient file and the laboratory parameters collected (renal function). The interaction analysis was supported by the interaction databases of MMI-PharmIndex^®^Plus, Vidal MMI Germany GmbH, Version 2020.4, and Lexicomp^®^ Drug interactions, 2020 UpToDate^®^, Wolters Kluwer.

In accordance with the recommendations of the International Group for Reducing Inappropriate Medication Use & Polypharamacy (IGRIMUP) [25], the medication of all the patients visited was analyzed according to the following parameters [26]:

Adverse drug reactions (ADRs) in general and their sub-categories:

Indication and drug selection:

- Overuse (missing prescription for existing indication). 

- Underuse (missing indication for prescribed drug).

- Inadequate care (existing indication but unfavorable drug prescription, e.g., contraindication, double prescription, and unsuitable drug and drug form).

Dosage:

- Incorrect or inadequate dose, incorrect dose interval, missing drug monitoring.

Application:

- Incorrect route of administration (oral, intravenous, transdermal, and others), incorrect preparation, and incorrect application duration.

- Interaction (drug–drug interaction = DDI): between two or more drugs.

Side effects:

Documentation of medical orders:

- Transmission errors in the patient chart or incorrect medical orders, e.g., the lack of potency in a drug.

All patient-related clinically relevant potential adverse drug reactions (ADRs) were recorded and assessed by the pharmacist, and a recommendation was made to avoid ADRs.

In the pharmaceutical-medical consultation, the possible benefits and potential harm due to adverse drug reactions were weighed against each other and were jointly decided in a professional dialog for the benefit of the patient. The personal priority of the patient concerned (the patient’s decision on his medication) or their relatives was also taken into account in the decision on how to proceed.

The outcome, whether the geriatrician implemented or rejected the pharmacist’s recommendation and whether it was a therapeutic drug monitoring (TDM) measure (such as a laboratory control, electrocardiogram (ECG) check, or patient monitoring for symptoms, side effects and/or effect alone) or a medication modification (such as drug onset, discontinuation, changes, dosage changes, and dosage form changes), was also recorded.

The present study now exclusively evaluates the antipsychotic prescribing patterns in younger and older geriatric patients with polypharmacy and the impact of pharmaceutical-medical counseling in the form of a weekly visit on antipsychotic use. Data analysis was performed in an anonymous form.

### 2.2. Study Population

A total of 744 patients were seen and discussed during the weekly pharmaceutical medical rounds from January 2018 to June 2020. (Figure 1) After excluding a total of 73 patients who did not meet the criteria of polypharmacy (defined as >5 prescriptions) (*n* = 24), aged 70 years (*n* = 27), or whose data were incomplete (*n* = 22), the remaining 671 geriatric patients were divided into two groups: with (*n* = 206) and without (*n* = 465) antipsychotics. Only patients with at least one antipsychotic were included in the analysis (*n* = 206) (30.7%). (Figure 1) In order to be able to study the prescribing patterns for antipsychotics and the influence of a pharmaceutical-medical visit in geriatric patients with polypharmacy of different ages, the group was stratified into two sub-groups: patients with antipsychotics aged 70 to 84 years, referred to in the study as the “young-old” (*n* = 113) and patients with antipsychotics aged 85 to 100 years, referred to here as the “old-old” (*n* = 93) [27].

### 2.3. Statistical Methods

Regarding the description of the study population, the absolute and relative frequencies have been given. Differences between the two study groups were tested using a chi-squared test or the Fisher exact test for categorical variables and for normally distributed continuous variables with the T-test or the Mann–Whitney U-test in case of skewed distributions. In order to explore the associations of the dependent variable ADRs and its sub-items with independent variables, such as age, gender, glomerular filtration efficiency, as well as prescribed drugs, binary logistic regression models were used. The analysis was performed with the total group (70–100 years) as well as the sub-groups: young-old patients (70–84 years) and old-old patients (85–100 years). A *p*-value < 0.05 was considered to be significant. All *p*-values are purely exploratory.

Statistical analyses were performed using IBM SPSS Statistics for Windows version 26.0, Armonk, New York, USA. Microsoft Excel^®^2019 was used for graphical mapping, and Microsoft Word^®^2019 was used for text and tables.

### 2.4. Ethics Consideration

Ethical approval was obtained from the Ethics Committee, University Bonn, Germany (No. 345/20). The Ethics Committee confirmed that no consultation was required for the retrospective evaluation of the data obtained in the course of routine diagnostics. There were no professional legal, or ethical concerns.

## 3. Results

The age of the two study groups differed significantly (*p* = < 0.001), with a mean age of 79.1 years (standard deviation (SD) of ±3.90 years) in the young-old study group and a mean age of 89.5 years (SD ±3.77 years) in the old-old group. There was no significant difference in the two study groups in terms of gender distribution, the presence of extreme polypharmacy (defined as the number of drugs >10 prescriptions), and the average number of drugs prescribed both before and after the pharmaceutical-medical consultation. The old-old study group had significantly higher renal insufficiency stages compared to the young-old group (*p* < 0.001). This is also reflected in the comparison of the higher renal insufficiency stages (<30 mL/min) (*p* = 0.03) (Table 1).

### 3.1. Prescribing Patterns

Prescription patterns of antipsychotics show that a total of 11 different antipsychotics were used in the geriatric ward, with the three risperidone, quetiapine, and pipamperone agents being prescribed most frequently. In the young-old group, quetiapine was prescribed most frequently, followed by pipamperone and risperidone; in the old-old group, pipamperone was prescribed most frequently, followed by risperidone and quetiapine. (Figure 2).

The overview of the use of antipsychotics, classified by their potency, also shows that there is little significant difference between the two age groups. Only the combination of Pipamperone and Risperidone was found significantly more often in the old-old age group. (Table 2).

Table 3 lists the usual dosages of each antipsychotic used in a geriatric facility, including the maximum dose for each.

### 3.2. Safety of Medications Used

Regarding antipsychotic use, the ADR’s analysis revealed that the most frequent conspicuous features in both the age groups concerning interactions with antipsychotics (16.8% (*n* = 19) in the young-old group vs. 11.8% (*n* = 11) in the old-old group), followed by anomalies on the indication and drug selection of antipsychotics (10.6% (*n* = 12) in the young-old group vs. 7.5% (*n* = 7) in the old-old group), in each case with no significant difference in the two age groups.

Only on the question of overuse of antipsychotics, as a sub-analysis of the indication and drug selection, was there a significant difference between the two age groups. The age group 70–84 years had a significantly (*p* = 0.04) higher use of antipsychotics with 7.1% (*n* = 8) than the age group 85–100 years with 1.1% (*n* = 1). (Table 4).

Associations between age, gender, glomerular filtration efficiency, the number of drugs prescribed, and the occurrence of ADRs and their sub-categories (= dependent variables) are shown in Table 5. The analysis was performed first with the entire study group (age 70–100 years), followed by the younger (age 70–84 years) and older (age 85–100 years) geriatric patient groups. Binary logistic regression analyses revealed a significant association between the total number of ADRs and the subcategory “interaction” (drug–drug interaction) for the entire study group. The more medications prescribed, the more likely adverse drug reactions or interactions occur. Increasing the medication by one drug increased the odds regarding the occurrence of ADRs or interactions by 1.10-fold, *p* = 0.02, and 1.12-fold, *p* = 0.02, respectively. The odds increase by 10% and 12%, respectively. The sub-analyses of the two age groups showed that these associations were mainly explained by the older study group. In addition, there was also a significant association between age and the occurrence of ADRs in the older study group. Increasing the medication in the old-old group by one drug increased the odds regarding the occurrence of ADRs or the interactions by 1.24-fold, *p* = 0.01 and 1.31-fold, *p* = 0.01, respectively. With increasing age in the older study group, the occurrence of ADRs was reduced by 0.80-fold, *p* = 0.04. The glomerular filtration efficiency also showed a borderline significant result regarding drug–drug interactions (1.04-fold, *p* = 0.04). The more limited the renal performance, the more frequent drug–drug interactions occurred. The younger study group additionally showed a correlation between age and overuse (1.38-fold, *p* = 0.04). The older the patients in this “young” study group were, the more frequently an overuse seemed to be observed.

### 3.3. Impact of a Pharmaceutical Medical Consolation

Regarding antipsychotics, a total of 55 recommendations (26.7%) were made by the pharmacist in the joint pharmaceutical-medical rounds. Most of these recommendations (*n* = 49) were implemented by geriatricians. Only six recommendations were rejected. In addition, the individual TDM or medication modification recommendations are also listed. Both overall (acceptance and rejection rate) as well as individual recommendations (such as laboratory control or drug discontinuation) yield no significant differences between the two age groups. (Table 6).

Important examples of potentially and clinically relevant ARDs, the resulting pharmaceutical recommendations, and their physician implementation that emerged during the pharmaceutical-medical dialogue are listed in Table 7.

## 4. Discussion

The aim of this study was to investigate the use of antipsychotics in terms of prescribing patterns, their safe use, and the impact of pharmaceutical-medical counseling in younger and older geriatric patients with polypharmacy.

Although slight differences were found in the frequency of the most commonly prescribed antipsychotics in the two age groups studied, there were few significant differences in the choice of antipsychotics prescribed between the two groups. With regard to drug safety and/or adverse drug reactions (ADR) to antipsychotics, only the sub-analysis showed a significant overuse in the group of young-old patients. The binary logistic regression analyses revealed that ADRs and DDIs were particularly related to the number of medications prescribed. The impact of a pharmaceutical-medical consultation shows that, despite high geriatric expertise in the treatment of older patients in a geriatric department, this is a significant need for counseling and support in a quarter of those concerned with a high rate of implementation of the recommendations given.

### 4.1. Prescribing Patterns

Antipsychotics are commonly used in older people [28]. In the available literature, the frequency of antipsychotic use is reported up to 45.1% [3,5,29]. In the present study, the proportion of patients with at least one antipsychotic in their medication was 30.7%. There is no comparison between young and old antipsychotic users in the literature so far. The frequency of antipsychotic prescriptions (at least one antipsychotic) varied little between the age groups: 30.8% in the young old group and 30.6% in the elderly old group. Because of an increasing rate of dementia with age [30], and thus, delirium [31,32], one might have expected a higher antipsychotic prescription rate in the older versus younger study group. This was not observed in our study analysis. However, antipsychotics, such as pipamperone and risperidone, are consistently used medications in the setting of delirium [16]. Unfortunately, a more precise examination of the indication (whether, for example, delirium was present) can no longer be clarified with the available routine data since only the recommendations, but not the underlying diseases, were documented in the database.

Moreover, there was little difference in the prescribing patterns between the two age groups with respect to neuroleptic potency. However, a higher prescription rate in the combination of pipamperone and risperidone was found in the older study group (*p* = 0.01). This may be due to an age-related higher rate of dementia/delirium in very old patients [30,32].

### 4.2. Safety of Medications Used

With regard to medication safety, differences were only found in the sub-analyses in the ADR “indication and drug selection” and in the sense of overuse in the young-old patients. In young-old patients, antipsychotics appear to be used rather more frequently without a confirmed indication. From a critical point of view, it is clear that in young-old patients, antipsychotics are more likely to be used without a reliable indication. In older-old patients, it seems that more caution is exercised—possibly in the knowledge of the risks of antipsychotics in old age. All other ADRs (underuse, inadequate care, dosage, application, interaction, and side effects) showed no significant differences between the age groups. 

Overall, the use of antipsychotics in older geriatric patients seems to be comparably safe in terms of underuse, inadequate care, dosage, application, interaction, side effects, and documentation errors as their use in younger geriatric patients. 

Antipsychotics show a variety of ADRs, but with no difference in the two studied age groups. In this respect, the use of antipsychotics in the hands of experienced therapists seems to be sufficiently safe to be practically used also in old-aged patients. Nevertheless, special caution and patient-specific risk-benefit assessment are always required when using antipsychotics.

A binary logistic regression analysis found a significant relationship between the occurrence of the dependent variable ADR and interactions with the independent variable number of medications prescribed. The higher the number of prescribed drugs, the higher the risk of ADRs and drug–drug interactions (DDI). Our results are, therefore, in line with the results of other authors who have also described a strong correlation between the number of prescribed drugs and the occurrence of ADRs or drug–drug interactions [11,33,34]. The risk of potential drug–drug interactions even seems to increase almost exponentially with the number of drugs used [34]. The consequence is that, whenever possible, adequate polypharmacy should be sought, and, if possible, deprescribing should be attempted [9,11]. Detailed analysis shows that the association between the number of medications prescribed and ADRs or DDIs applies primarily to the older study group. In clinical practice, this means that special caution is required, especially for very old patients.

### 4.3. Impact of a Pharmaceutical Medical Consolation

The present study shows that a pharmaceutical-medical dialogue, here as a weekly pharmaceutical-medical consultation, supports the identification of patient-specific potential ADRs of antipsychotics in geriatric patients with polypharmacy. Thus, potential ADRs were detected in the areas of indication and drug selection, dosage, application, interaction, side effects, and documentation.

In recent years, it has been repeatedly shown that the involvement of a pharmacist in clinical routines has had a positive impact on therapy and therapy safety.

The effect of pharmaceutical-medicinal consultation in a geriatric setting has also been investigated in several studies [35,36,37,38]. Lee et al. [37] found in their systematic review and meta-analysis that the integration of a pharmacist into clinical practice led to an improvement in therapy, safety, hospitalization, and adherence; however, the results are based on patients cared for primarily in the ambulatory setting.

Kiesel et al. [36] systematically reviewed the impact of clinical pharmacists in geriatric hospital care in Europe and similarly concluded that pharmacists can increase their medication appropriateness and drug safety in geriatric patients, often resulting in concomitant cost savings. Furthermore, Rösler et al. [38] reported an improvement in the medication management of geriatric patients following a yearly trial of integrating a pharmacist into the multidisciplinary team of a geriatric department.

To the author’s knowledge, no study has investigated whether there are differences in the impact between young-old and old-old patients in the heterogeneous group of geriatric patients. In this study, a potential ADR was detected by the pharmacist in a total of 26.7% of geriatric patients with antipsychotic polypharmacy, and recommendations for avoidance were suggested. The proportion of geriatric patients with antipsychotics who seemed to benefit from a pharmaceutical-medical dialogue was thus high. There was no significant difference between the two age groups in the recommendations for antipsychotic use in the context of TDM and/or medication modifications, nor in the acceptance or rejection of recommendations. Sub-category analysis also showed no significant differences between the two age groups studied. Almost all recommendations were implemented by the geriatricians. Regarding the use of antipsychotics, this study confirmed the positive effect of the support of the interdisciplinary team by a pharmacist [36,37,38] and this in both young and old age groups with polypharmacy. When including geriatric-specialized departments, the pharmacist assists in the recognition and prevention of adverse effects and medication errors in the use of antipsychotics in all geriatric patients.

### 4.4. Strengths and Weaknesses

To the best of our knowledge, this is the first time that prescriptions of antipsychotics in younger and older geriatric patients with polypharmacy, their safety, and the impact of a pharmaceutical-medical dialogue on antipsychotic use have been evaluated. The strengths of the current study include a large patient population and the use of real-world clinical data.

However, our study has some limitations which should not be left unmentioned. Due to the retrospective nature of the study, possible causal relationships can be suggested but cannot be definitively proven. Before the evaluation, all the collected data were anonymized so that it was no longer possible to refer back to the individual patient and his or her patient file at this point in time. This makes it impossible to subsequently match data that was not collected, such as liver values. Among other things, this also affects the information regarding ADRs. Although the type of ADR and resulting recommendations have been documented, the indications for the drugs used have not. It is, therefore, no longer possible to determine on the basis of which indicates certain drugs were administered. The study shows the optimization of medication without being able to make a concrete statement about the adverse drug reactions actually avoided as a result. Nor is it possible to determine the exact cause of the significant difference in the overuse observed in the young and old age groups. Similarly, no information on liver dysfunction or comorbidities was documented during the weekly rounds. However, polypharmacy can be seen as a surrogate factor of multimorbidity/comorbidity [39,40]. The comorbidity increases with the number of medications taken [41,42].

It is known from the published literature that up to 40% of people over the age of 60 take over-the-counter nutritional supplements [41]. In our evaluation, these supplements can be discounted because they were not continued or prescribed during the inpatient stay.

The data were collected and recorded with no known subsequent additional use by a retrospective evaluation. Thus, negative influences/biases can be excluded based solely on the nature of the data collected.

The study was conducted in a designated geriatric center that has extensive experience in the treatment of older people. This may have led to a certain selection bias. The transferability of the results to a target population of “elderly patients in a non-geriatric clinic” who do not have this special experience may be limited (= external validity). Furthermore, the study is a monocentric evaluation. Therefore, a transfer to other centers is not possible without further consideration.

## 5. Conclusions

The prescriptions of antipsychotics in geriatric patients with polypharmacy, their safety, and the impact of a pharmaceutical-medical dialogue on the use of antipsychotics seem comparable between younger and older geriatric patients in the geriatric setting. Antipsychotics should always be critically considered and be used cautiously and judiciously after a patient-specific risk-benefit- assessment, re-evaluated regularly for continued indication, and prescribed optimally in the dialogue between geriatrician and pharmacist to avoid potential ADRs. The number of drugs prescribed seems to have a decisive influence on the occurrence of ADRs and drug–drug interactions. The more drugs are used, the higher the risk of ADRs. Regular pharmaceutical-medical dialogues, for example, in the form of a pharmaceutical-medical consultation, are recommended for all geriatric patients with polypharmacy and while taking antipsychotics.

## Figures and Tables

**Figure 1 biomedicines-10-03127-f001:**
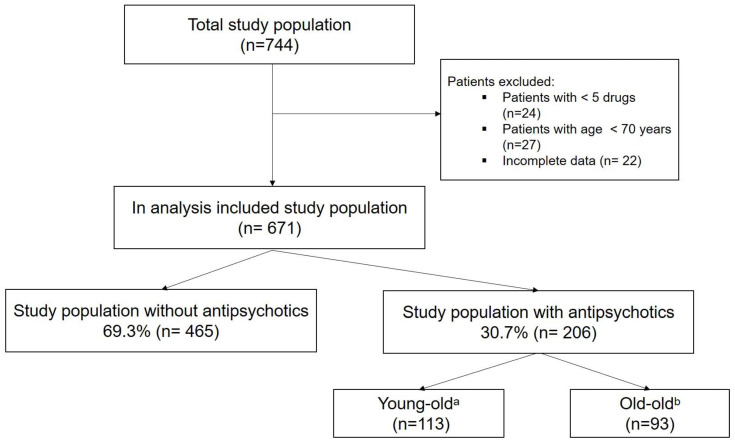
Study population. ^a^ 70–84 years. ^b^ 85–100 years.

**Figure 2 biomedicines-10-03127-f002:**
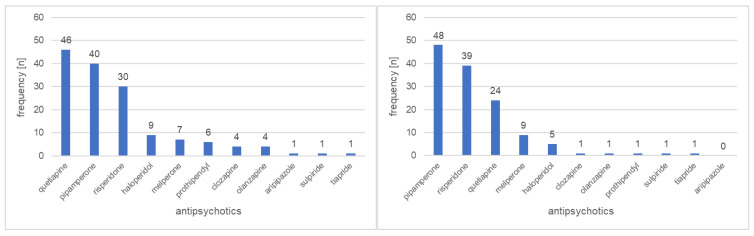
Young-old group; aged 74–84 years (*n* = 113). Old-old group; aged 85–100 years (*n* = 93).

**Table 1 biomedicines-10-03127-t001:** Characteristics of study populations.

	Age 70–84 *n* = 113 (%)	Age 85–100 *n* = 93 (%)	*p*-Value
**Age, years (mean ± SD)**	79 ± 4	90 ± 4	**<0.001** T-Test
**Female gender**	69 (61.1%)	51 (54.8%)	0.37
**Polypharmacy (>10 drugs)**	90 (79.6%)	68 (73.1%)	0.27
**Number of drugs per patient**			
**Before recommendation ^a^**	14 ± 4	13 ± 4	0.30 T-Test
**After recommendation ^a^**	14 ± 4	13 ± 4	0.47 T-Test
**Renal insufficiency stages**			**<0.001**
**stage 1**	37 (32.7%)	9 (9.7%)	
**stage 2**	40 (35.4%)	31 (33.3%)	
**stage 3**	33 (29.2%)	44 (47.3%)	
**stage 4**	3 (2.7%)	9 (9.7%)	
**stage 5**	0 (0.0%)	0 (0.0%)	
**Renal glomerular filtration rate**			**0.03**
**>30 mL/min (stage ≤3)**	110 (97.3%)	84 (90.3%)	
**≤30 mL/min (stage ≥4)**	3 (2.7%)	9 (9.7%)	

^a^ Recommendation here means all clinically relevant recommendations mentioned by the pharmacist, not exclusively antipsychotics. Renal insufficiency stages: stage 1: >89 mL/min, stage 2: 60–89 mL/min, stage 3: 30–59 mL/min, stage 4: 15–29 mL/min, and stage 5: <15 mL/min. Significant results (*p* < 0.05) appear in bold letters. Unless otherwise stated, the chi-squared test was performed.

**Table 2 biomedicines-10-03127-t002:** The use of antipsychotics in the two age groups classified according to their neuroleptic potency.

	Age 70–84 *n* = 113 (%)	Age 85–100 *n* = 93 (%)	*p*-Value
**Antipsychotics**			
**High potency (monotherapy)**	18 (15.9%)	11 (11.8%)	0.40
Aripiprazole	0 (0%)	0 (0%)	
Haloperidol	5 (4.4%)	2 (2.2%)	0.46 *
Olanzapine	3 (2.7%)	1 (1.1%)	0.63 *
Risperidone	10 (8.8%)	8 (8.6%)	0.95
**Medium potency (monotherapy)**	38 (33.6%)	25 (26.9%)	0.30
Clozapine	3 (2.7%)	1 (1.1%)	0.63 *
Quetiapine	34 (30.1%)	22 (23.7%)	0.30
Sulpiride	1 (0.9%)	1 (1.1%)	1.00 *
Tiapride	0 (0%)	1 (1.1%)	0.45 *
**Low potency (monotherapy)**	23 (20.4%)	22 (23.7%)	0.57
Melperone	3 (2.7%)	6 (6.5%)	0.31 *
Pipamperone	16 (14.2%)	16 (17.2%)	0.57
Prothipendyl	4 (3.5%)	0 (0%)	0.13 *
**Combination of different** **Potencies ^a^**	34 (30.1%)	35 (37.6%)	0.25
Pipamperone + Risperidone	17 (15.0%)	28 (30.1%)	0.01
Pipamperone + Quetiapine	4 (3.5%)	1 (1.1%)	0.38 *
Other combinations ^b^	13 (11.5%)	6 (6.5%)	0.21

^a^ Combinations of antipsychotics, e.g., high-potency with low-potency, high-potency with medium-potency, medium-potency with low-potency antipsychotics. ^b^ For example, Melperone + Quetiapine or Quetiapine + Prothipendyl. The chi-squared test was performed. * Fisher-test was used. Significant results (*p* < 0.05) appear in bold letters.

**Table 3 biomedicines-10-03127-t003:** Standardized dosing regimens of the most commonly used antipsychotics in the study population.

Antipsychotic Drug	Indication	Loading Dose ^a^	Increase/Maximum Dose ^a^
**Haloperidol**	Delirium in dementia ^a^ (2nd choice)	Loading dose 0.5 mg (oral)	If necessary, gradually increase up to max 5 mg/day in 2 doses.
**Melperone**	Sleeping disorders, psychomotor agitation	Gradual dosing with 25 mg (oral) at night	If necessary, increase in 25 mg increments. Max. 150 mg/day in 2–4 doses.
**Pipamperone**	Sleeping disorders, psychomotor agitation	Gradual dosing with 20 mg (oral) at night	If necessary, increase dose to 40 mg single dose, Max. three times daily (=120 mg/day)
**Quetiapine**	Psychotic symptoms in Parkinson’s disease	Gradual dosing with 12.5 mg (oral)	If response is inadequate, increase dose to 25 mg, then further increase in 25 mg increments if necessary. Max. 150 mg/day in 2 doses.
**Risperidone**	Delir in dementia^a^ (1st choice)	Gradual dosing with 2 × 0.25 mg/day (oral)	If response is inadequate, increase dose by 0.25 mg increments to up to 1 mg twice daily. For the majority of patients, the optimal dose is 2 × 0.5 mg/day.

^a^ After failure of non-pharmacological therapies and if there is a risk of danger to self or others.

**Table 4 biomedicines-10-03127-t004:** Differences in potential ADRs^a^ of antipsychotics in the two age groups by classification of pharmaceutical interventions.

	Age 70–84 *n* = 113 (%)	Age 85–100 *n* = 93 (%)	*p*-Value
**Indication and drug selection**	12 (10.6%)	7 (7.5%)	0.45
**Overuse ^b^**	8 (7.1%)	1 (1.1%)	**0.04 ***
**Underuse ^b^**	0 (0%)	0 (0%)	
**Inadequate care ^b^**	4 (3.5%)	6 (6.5%)	0.35 *
**Dosage ^c^**	4 (3.5%)	1 (1.1%)	0.38 *
**Application ^d^**	2 (1.8%)	0 (0%)	0.50 *
**Interaction ^e^**	19 (16.8%)	11 (11.8%)	0.33
**Side effects ^f^**	0 (0%)	1 (1.1%)	0.45 *
**Drug documentation error**	2 (1.8%)	3 (3.2%)	0.66 *

^a^ ADRs = adverse drug reactions ^b^ Overuse (missing prescription for existing indication) and underuse (missing indication for prescribed drug) as well as inadequate care (existing indication but unfavorable drug, e.g., contraindication, double prescription, and unsuitable drug and drug form). ^c^ Incorrect or inadequate dose, incorrect dose interval, and missing drug monitoring. ^d^ Type and duration. ^e^ Between two or more drugs. ^f^ Side effect as a direct consequence of the antipsychotic drug. Significant results (*p* < 0.05) appear in bold letters. The chi-squared test was performed. * Fisher-test was used.

**Table 5 biomedicines-10-03127-t005:** Correlations between dependent variables, such as ADRs (adverse drug reactions), and their subcategories and independent variables, such as age, gender, glomerular filtration efficiency (GFR) as well as the number of prescribed drugs in binary logistic regression models ^a^.

**Dependent Variables**																			
**Total study** **population (*n* = 206)**	**Independent** **variables**	**Total ADRs ^b^**		**ADR Sub-Items:**															
				**Indication/Drug Selection**		**Overuse**		**Inadequate Care**		**Dosage**		**Application**		**Interaction ^c^**		**Side Effects**		**Drug** **Documentation Error**	
		**adjOR**	**95% CI**	**adjOR**	**95% CI**	**adjOR**	**95% CI**	**adjOR**	**95% CI**	**adjOR**	**95% CI**	**adjOR**	**95% CI**	**adjOR**	**95% CI**	**adjOR**	**95% CI**	**adjOR**	**95% CI**
	**Gender**	0.98	0.51-.1.89	1.04	0.40–2.74	0.70	0.17–2.90	d		0.27	0.03–2.76	1.62	0.08–31.37	1.03	0.46–2.33	d		2.09	0.34–12.89
	**Age**	0.95	0.90–1.00	1.01	0.94–1.10	0.98	0.88–1.09	d		0.85	0.72–1.01	0.89	0.66–1.20	0.95	0.89–1.02	d		1.01	0.87–1.18
	**GFR**	1.01	1.00–1.03	1.02	0.99–1.04	1.01	0.97–1.04	d		1.00	0.96–1.05	d		1.02	1.00–1.04	d		0.98	0.94–1.03
	**Number of drugs**	**1.10**	**1.02–1.20**	1.03	0.91–1.16	0.95	0.80–1.14	d		1.16	0.93–1.46	0.88	0.55–1.43	**1.12**	**1.02–1.24**	d		0.97	0.78–1.21
**Age 70–84 (*n* = 113)**	**Gender**	0.82	0.35–1.90	0.59	0.14–2.43	0.65	0.11–3.75	0.38	0.03–4.37	0.40	0.04–4.50	2.12	0.09–51.77	0.83	0.29–2.38	d		0.78	0.03–19.67
	**Age**	0.97	0.87–1.07	1.21	0.99–1.48	**1.38**	**1.02–1.89**	1.00	0.76–1.31	0.81	0.63–1.05	0.96	0.65–1.41	0.92	0.81–1.04	d		1.11	0.71–1.73
	**GFR**	1.01	0.99–1.03	1.02	0.99–1.05	1.01	0.97–1.05	1.06	0.98–1.14	1.02	0.96–1.08	d		1.01	0.98–1.03	d		0.94	0.87–1.03
	**Number of drugs**	1.05	0.95–1.17	1.00	0.84–1.20	0.89	0.70–1.13	1.23	0.92–1.63	1.13	0.86–1.48	0.85	0.51–1.42	1.06	0.93–1.20	d		0.96	0.65–1.40
**Age 85–100 (*n* = 93)**	**Gender**	1.28	0.42–3.93	2.95	0.53–16.44	d		2.31	0.39–13.74	d		d		1.55	0.36–6.62	d		2.10	0.18–24.74
	**Age**	**0.80**	**0.65–0.99**	0.88	0.66–1.16	d		0.86	0.62–1.18	d		d		0.79	0.60–1.06	d		0.86	0.55–1.33
	**GFR**	1.03	1.00–1.06	1.00	0.96–1.04	d		1.02	0.97–1.06	d		d		**1.04**	**1.00–1.08**	d		1.01	0.95–1.07
	**Number of drugs**	**1.24**	**1.05–1.47**	1.06	0.87–1.28	d		1.06	0.87–1.31	d		d		**1.31**	**1.07–1.61**	d		0.99	0.73–1.33

adjOR = adjusted odds ratio; 95% CI = 95% confidence interval. Significant results are highlighted. ^a^ Models were adjusted for age, gender, GFR, and the number of prescribed drugs. ^b^ ADRs = adverse drug reactions as dependent variable. ^c^ Interaction = drug–drug interaction (DDI). ^d^ OR cannot be estimated. Significant results (*p* < 0.05) appear in bold letters.

**Table 6 biomedicines-10-03127-t006:** The number of patients in both age groups with accepted or rejected pharmaceutical recommendations for antipsychotic use, divided into therapeutic drug monitoring (TDM) and medication modifications.

	Age 70–84 *n* = 113 (%)	Age 85–100 *n* = 93 (%)	*p*-Value
**TDM recommendations ^a^**			
**Accepted**	10 (8.8%)	6 (6.5%)	0.52 *
Laboratory control	0 (0.0%)	2 (2.2%)	0.20
ECG check	6 (5.3%)	2 (2.2%)	0.30
Patient monitoring for symptoms, side effects, and/or effect alone	4 (3.5%)	2 (2.2%)	0.69
**Rejected**	2 (1.8%)	0 (0.0%)	0.50
**No recommendation**	101 (89.4%)	87 (93.5%)	0.29 *
**Medication modification recommendations**			
**Accepted**	20 (17.7%)	13 (14.0%)	0.47 *
Drug onset	0 (0.0%)	1 (1.1.%)	0.45
Drug discontinuation	16 (14.2%)	6 (6.5%)	0.08 *
Drug changes	0 (0.0%)	2 (2.2%)	0.20
Dosage changes	2 (1.8%)	4 (4.3%)	0.41
Dosage form changes	2 (1.8%)	0 (0.0%)	0.50
**Rejected**	4 (3.5%)	0 (0.0%)	0.13
**No recommendation**	89 (78.8%)	80 (86.0%)	0.18 *

^a^ Therapeutic drug monitoring (TDM). Significant results (*p* < 0.05) are highlighted. Unless otherwise stated, the Fisher-test was performed. * The chi-squared test was performed.

**Table 7 biomedicines-10-03127-t007:** Examples of potential adverse drug reactions, the pharmaceutical intervention recommendation, and the geriatrician’s measure.

Active Agent/s	Adverse Drug Reaction	Pharmaceutical Intervention Recommendation	Geriatrician’s Measure
Carbamazepine and quetipaine	CYP3A4- interaction: Loss of effect of quetiapine may occur.	Risk-benefit consideration of carbamazepine due to high side effect potential and interactions. Tapering of carbamazepine.	Tapering of carbamazepine (with concomitant administration of other anticonvulsants).
Quetiapine sustained-release tablets	Quetiapine sustained-release tablets are mortared as a non-mortar dosage form. Loss of the retarded effect.	Change to unretarded dosage form with examination and, if necessary, adjustment of the dose interval.	Changeover to unretarded dosage form occurs.
Quetiapine and citalopram	Risk of QT prolongation with arrhythmia.	ECG * and electrolyte controls with drug combination.	Close ECG and electrolyte controls were ordered.
Quinagolide and pipamperone	Dopamin-Agonist vs. Dopamin- Antagonist. Mutual loss of effect possible.	Deprescribing of pipamperone recommended.	Pipamperone discontinued and patient observed for the need of further drug therapy.

* Electrocardiogram.

## Data Availability

The data that support the findings of the study are available from the corresponding author A.L., upon reasonable request.

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
