# Peer review of "Prescriptions of Antipsychotics in Younger and Older Geriatric Patients with Polypharmacy, Their Safety, and the Impact of a Pharmaceutical-Medical Dialogue on Antipsychotic Use"

_biomedicines, 2022, doi:10.3390/biomedicines10123127_

Round 1
Reviewer 1 Report
Dear Authors,
I have read with interest this manucript and now I send you my comments:
1) the results must be imporved, it is important to add data related to the characteristic of the patients in order to evaluate in the ADRs are or not related to renal or lievr failure. Moreover, it is important report data of ADRs and DDIs ineach patients to evaluate the role of gender, age, comorbidity and poly-therapy. Moreover, please add also data related to supplements use and possible role in the development of ADRs.
2) Figures and tables must be deleted and changed with more specific tables and fures showing the use of drug and not class of drugs in each patients also consideriing the dosage.
3) In table you report the TDM, please add data of TDM and measure that has been used after these data
4) Discussion must be rewritten considering these data
5) These data are necessary for the acceptation of the manuscript
Reviewer 2 Report
This paper entitled "Prescriptions of antipsychotics in younger and older geriatric patients with polypharmacy, their safety, and the impact of a pharmaceutical-medical dialogue on antipsychotic use" has been evaluated prescriptions of antipsychotics in younger and older geriatric patients with polypharmacy, their safety, and the impact of a pharmaceutical-medical dialogue on antipsychotic use. These will be helpful in some degree for the use of antipsychotics. In general, this paper is well designed and written. In my view, this paper could be published in Biomedicines without further revisions.
Author Response
Thank you for the kind feedback and good review.
No changes required.
Reviewer 3 Report
The subject of the article is up-to-date and the study is well designed.
There are some typos and it is appropriate to correct them.
Author Response
Thank you for the kind feedback and good review.
Reviewer 4 Report
It is a retrospective analysis of the outcome of weekly geriatric patients' medical-pharmaceutical consultations done over 30 months. Of the total 744 patients discussed, 671 were the ones with polypharmacy, of which roughly 1/3 (206) were on antipsychotics. And these last were the authors' interests and concerns. The study points to the importance of geriatrician-pharmacist dialog in prescription and the follow-up of antipsychotic drugs to avoid adverse reactions, to carefully assess risk-benefit, and increase the quality of life in geriatric patients with polypharmacy.
Author Response

(The authors gave the same response as above.)

Round 2
Reviewer 1 Report
Dear Authors,
I have read the manuscript and I think that it has been improved but it is not a clinical study but an epidemiological study with few clinical data.
Author Response
Dear Reviewer,
thank you for the feedback. We also think that through your suggestions the article could be improved.
The paper shows a practical clinical application of a joint visit of a pharmacologist with the treating physicians. The data collected are from the clinical practice of elderly geriatric patients. Possible recommendations by the pharmacologist and their degree of implementation is presented for the two study groups. We find that the work has a strong relation to clinical practice, because it also originated from practice.